# Mitochondrial Role in Oncogenesis and Potential Chemotherapeutic Strategy of Mitochondrial Infusion in Breast Cancer

**DOI:** 10.3390/ijms232112993

**Published:** 2022-10-27

**Authors:** Prisha S. Patel, Christopher Castelow, Disha S. Patel, Syamal K. Bhattacharya, Cem Kuscu, Canan Kuscu, Liza Makowski, James D. Eason, Amandeep Bajwa

**Affiliations:** 1Department of Surgery, Transplant Research Institute, James D. Eason Transplant Institute, College of Medicine, The University of Tennessee Health Science Center, Memphis, TN 38103, USA; 2School of Interdisciplinary Studies and Global Education, Belmont University, Nashville, TN 37212, USA; 3Department of Medicine, Division of Cardiovascular Diseases, College of Medicine, The University of Tennessee Health Science Center, Memphis, TN 38103, USA; 4Department of Microbiology, Immunology, and Biochemistry, College of Medicine, The University of Tennessee Health Science Center, Memphis, TN 38103, USA; 5Department of Medicine, Division of Hematology and Oncology, College of Medicine, The University of Tennessee Health Science Center, Memphis, TN 38103, USA; 6Department of Genetics, Genomics and Informatics, College of Medicine, The University of Tennessee Health Science Center, Memphis, TN 38103, USA

**Keywords:** cancer, mitochondria, mitotherapy, triple negative breast cancer, metabolism

## Abstract

Triple negative breast cancer (TNBC) is one of the most aggressive cancers diagnosed amongst women with a high rate of treatment failure and a poor prognosis. Mitochondria have been found to be key players in oncogenesis and tumor progression by mechanisms such as altered metabolism, reactive oxygen species (ROS) production and evasion of apoptosis. Therefore, mitochondrial infusion is an area of interest for cancer treatment. Studies in vitro and in vivo demonstrate mitochondrial-mediated reduction in glycolysis, enhancement of oxidative phosphorylation (OXPHOS), reduction in proliferation, and an enhancement of apoptosis as effective anti-tumor therapies. This review focuses on mitochondrial dysregulation and infusion in malignancies, such as TNBC.

## 1. Introduction

### 1.1. Epidemiology and Hallmarks of Cancer

Breast cancer is the most prominent malignancy diagnosed in women, and the incidence is expected to continually increase over 250,000 in 2021 and approximately 364,000 by 2040 [1,2]. Triple negative breast cancer (TNBC) is one of the most aggressive subtypes of breast cancer with poor response to conventional therapy due to a lack of estrogen receptors (ER), progesterone receptors (PR), and human epidermal growth factor receptor 2 (HER2). TNBC accounts for approximately 12% of breast cancer cases in the United States. Compared to breast cancer positive for hormone receptors, TNBC has up to a 16% reduced 5-year survival, making better treatment options essential [3].

Hanahan and Weinberg differentiated cancer cells from non-cancer cells by defining necessary “Hallmarks of Cancer” including sustained proliferation, evasion of growth suppression, immortalization, invasion & metastasis, angiogenesis, and apoptosis [4]. Additionally, the article described other emerging hallmarks such as altered metabolism and immune evasion, as well as enabling characteristics including tumor promoting inflammation and genetic instability & mutations [4]. Cancer cells undergo many transformations, such as increased dependence on glycolysis and evasion of apoptosis; many of these functions occur due to alterations in the cancer cell mitochondria metabolism. Thus, mitochondrial function impacts energy metabolism, calcium homeostasis, and apoptosis as well as impacting innate immunity and stem cell regulation in cancer and non-cancer cells. Making mitochondrial targeted therapies including mitochondrial therapy (mitotherapy) or other pharmacological interventions potential areas of interest for treating cancer. This review highlights the impact of mitochondria on oncogenesis and the prognosis of mitochondrial therapy as a novel intervention, especially in TNBC.

### 1.2. Mitochondria: The Powerhouse

Mitochondria provide the cell with energy in the form of adenosine triphosphate (ATP). Mitochondria have cristae to increase surface area, so there is more space for ATP production. Healthy non-cancerous cells undergo oxidative phosphorylation (OXPHOS), which is a much more effective way to obtain ATP compared to glycolysis as oxygen is reduced to water via the proton gradient. Altering the electron transport chain (ETC) can cause a decrease in oxidative phosphorylation, and hence, ATP production. However, cancer cells are less reliant on OXPHOS because of the Warburg Effect. Cancer cells switch to aerobic glycolysis and lactic acid fermentation for energy production in response to hypoxia and/or nutrient deprivation; this, in turn, generates more reactive oxygen species (ROS). On the other hand, the Reverse Warburg Effect has been described as surrounding stromal cells supplying glycolytic products, such as ATP, for cancerous cells to drive proliferation and apoptotic resistance [5]. In lymph nodes with metastatic breast cancer, an increase in mitochondria and OXPHOS was noted in cancerous cells while stromal cells showed increased glycolysis [6]. Lack of caveolin-1 (Cav-1) signifies the Reverse Warburg Effect; for example, in the stroma of human breast cancer tissue, the lack of Cav-1 was correlated to recurrence and metastasis [7]. TNBC has been shown to be more resilient against glycolytic insults compared to hormonally responsive tumors. When given iodoacetate for glycolytic inhibition, TNBC showed increased numbers of active mitochondria, lower apoptotic factors of Bcl2 associated X (BAX), Bcl2-homologous antagonist killer (BAK), and caspase 9, lower ROS, and increased levels of p21 and poly [ADP-ribose] polymerase 1 [8].

Unlike many other organelles, the mitochondria have their own genomes, which encode for nuclear DNA (nDNA) and mitochondrial DNA (mtDNA). Many patients fail therapy because tumors develop resistance to multiple therapies, which may be due to nDNA and/or mtDNA mutations induced by certain therapies. The variety and the severity of mtDNA mutations play a role in tumorigenesis; for example, Complex I of the ETC is more likely to be mutated in cancerous phenotypes [9]. Complex I creates a proton gradient by oxidizing nicotinamide adenine dinucleotide (NADH) and reducing ubiquinone to ubiquinol; this step of OXPHOS produces ROS. The generation of ROS has been shown to cause mtDNA mutations in mice [10]. OXPHOS protein levels, specifically NADH:Ubiquinone Oxidoreductase Complex Assembly Factor 8 (NDUFB8), succinate dehydrogenase B (SDHB), and ATP synthasome proteins have been shown to be decreased in MDA-MB-231 (human TNBC cell line) when compared to MCF7 (human breast cancer cell line) [11]. Addition of NADH dehydrogenase subunit 1 from yeast (NDi1), a part of Complex 1, to human TNBC cell lines decreased metastasis and proliferation [12]. NDi1 is a part of Complex I, so. On the other hand, there was an increase in Complex IV activity in human TNBC cell lines [11]. The preceding studies emphasize the importance of mitochondria and OXPHOS, especially Complex I, in cancer cells. Even if SDH is mutated, cancer cells still undergo mitochondrial metabolism by producing ROS and metabolites for the Krebs Cycle [13]. For example, there was an increase in hypoxia-inducing factor alpha (HIFα) and ROS even when succinate dehydrogenase B (SDHB) was lost [14].

### 1.3. Mitochondria’s Role in Oncogenesis

Over the years, mitochondria have been found to play a key role in oncogenesis, tumor progression, and even metastasis. Porporato et al. described three major mechanisms by which mitochondria contribute to tumor formation including the production of ROS and the accumulation of oncometabolites and to apoptotic resistance via the disruption of the mitochondrial outer membrane protein (MOMP) or mitochondrial permeability transition (MPT) [15]. First, the production of ROS by mitochondria can initiate DNA damage and oncogenic signaling pathways. ROS has been shown to inactivate phosphatase and tensin homolog (PTEN); thus, PTEN cannot inactivate the PI3K/AKT pathway leading to cancer progression [16]. The need for ROS in tumor progression was illustrated with Trp53−/− mice, which had a survival advantage when kept in hypoxic conditions compared to Trp53−/− mice kept in standard atmospheric conditions (10 and 21% oxygen, respectively) [17]. Injecting mice with hydrogen peroxide has also been shown to enhance metastasis in carcinoma and subpopulations of breast cancer with higher levels of ROS and has also been associated with increased and earlier metastasis [18]. One reason for this increased metastatic potential is that ROS both increase matrix metalloproteinases (MMPs) and inactivate tissue inhibitors of metalloproteases (TIMPs) leading to invasion and metastasis. Additionally, endogenous ROS expression in tumors has shown to have a role in angiogenesis via reduction in micro-vessel density when a murine breast cancer model was treated with ortho-tetrakis-*N*-ethyl pyridyl porphyrin, a ROS scavenger [18]. On the other hand, a large and disproportional rise in ROS can induce cell cycle arrest and apoptosis through the release of cytochrome c from the mitochondria and the activation of caspases. They can also cause non-specific damage to macromolecules such as DNA, proteins, and lipids. Hydroxyl radicals are highly diffusible, which allows them to attack DNA by oxidizing DNA bases and causing single-stranded and double-stranded DNA breaks. ROS also modifies amino acid residues resulting in altered protein function and causing lipid peroxidation. The recent discovery of ferroptosis, a new type of regulated cell death (RCD) driven by iron-dependent lipid peroxidation, is another metabolic process by which ROS halt tumor progression. This occurs when the levels of ROS exceed the antioxidant activity inside of cells leading to the collapse of cellular homeostasis. This induction of ferroptosis then causes mitochondrial fragmentation and alterations in mitochondrial membrane potential [19]. The cellular damage induced by ROS has been used by several chemotherapeutics, such as cisplatin, to cause irreparable cellular damage and tumor cell apoptosis. However, when nuclear related factor 2 (Nrf2) was overexpressed in MCF7 cells, the cells were resistant to Cisplatin [20]. Nrf2 is a transcription factor that regulates the transcription of several antioxidant enzymes, such as NAD(P)H quinine oxidoreductase 1 (NQO1) [20]. Due to mitochondrial DNA susceptibility to DNA damaging agents, cell death can be induced by mutating mitochondrial DNA and raising ROS to a toxic level.

Lymphocytes are recruited to reduce ROS-induced damage, leading to inflammation. Apoptosis can also occur if repairs cannot be made, but if there is a BRCA mutation as with some breast cancers, apoptosis may not occur. BRCA1 deficiency causes an increase in ROS, which leads to inflammasome activation in mice and MCF7 cell lines [21]. When an inflammasome is activated in a BRCA1 deficient mouse with a tumor, macrophages proliferate and secrete cytokines that stimulate the production of M2 macrophages, especially in hypoxic regions to promote angiogenesis [22]. M2 macrophages are immunosuppressive and inhibit the activity of cytotoxic T cells, establishing a tumor-associated immunosuppressed microenvironment via Granulocyte-Macrophage colony-stimulating factor (GM-CSF) [21]. These M2 macrophages secrete TGF-ꞵ1, which along with other cytokines, switch the Suppressor of Mothers Against Decapentaplegic (SMAD) signaling growth suppressing pathway to the AKT signaling pathway that promotes tumor progression and aggressiveness [23]. The immunosuppressive microenvironment created by these M2 macrophages is favorable for tumor progression and eventual metastasis. These points illustrate that a careful balance of ROS production is necessary for tumor growth and progression.

The second mechanism by which mitochondria contribute to oncogenesis is by the accumulation of specific mitochondrial metabolites known as oncometabolites [16]. These oncometabolites include fumarate, succinate and 2-hydroxygluterate (2-HG) [24]. Some studies have found that the accumulation of these metabolites is sufficient to drive malignant transformation [16]. They do this by inhibiting different α-ketoglutarate dependent enzymes which lead to the inhibition of mitochondrial ATP synthase and causing expression of potentially oncogenic transcription factors [15]. The accumulation of fumarate can cause a process called succination, a non-enzymatic post-translational modification when performed on kelch like ECH-associated protein 1 can active transcription factor nuclear factor, erythroid derived 2 promoting oncogenesis [15].

The third mechanism is alterations in MOMP and/or MPT. These alterations in mitochondrial function help tumor cells avoid RCD that would typically halt progression of the neoplasm [15]. During apoptosis, the mitochondria release various proteins, including cytochrome c, Smac/DIABLO, and Omi/HtrA2. Smac/DIABLO and cytochrome c play a role in producing ATP and in activating caspases for apoptosis. Some chemotherapies induce apoptosis in cancerous cells via the intrinsic apoptotic pathway, in which cytochrome c is released from the mitochondria; this leads to the activation of caspases. If these genes are mutated, apoptosis cannot occur, and chemotherapy fails. Omi/HtrA2 plays a role in inhibiting antiapoptotic proteins, such as B-cell lymphoma 2 (BCL2). Cancer cells evade apoptosis by increasing BCL2 expression and promoting proliferation even in the absence of growth factors, such as epidermal growth factor (EGF), by promoting mitochondrial resistance to cell membrane permeabilization [15]. In addition, other oncogenes such as MYC and KRAS can alter mitochondrial dynamics and increase resistance to MOMP [25]. Some tumors have also been found to have an increased mitochondrial transmembrane potential that provides resistance to RCD. Alterations in mitochondrial metabolism and the maintenance of antioxidant defenses in tumor cells have been shown to be important in evading the ROS-driven MTP process by halting cytochrome-c mediated apoptosis [16].

### 1.4. Mitochondrial Infusion

Mitochondrial infusions may seem like a medicinal approach, but natural infusions have been documented by many individuals. Researchers have shown that cells can uptake mitochondria or mtDNA from surrounding cells without the addition of exogenous addition of mitochondria. Three uptake methods have been described: extracellular vesicles, tunneling nanotubes, and gap junctions. For extracellular vesicles, contents are packaged inside of a vesicle in the donor cell and released via exocytosis; the vesicle is then endocytosed into the recipient cell [26]. Tunneling nanotubes are formed via extension of the cell membrane of two cells, leading to bridging and transfer of content [27]. In gap junctions, connexin regulates what comes in and out of the cell; thus, regulating contents [27].

Since mitochondria dysfunction has been shown to cause tumorigenesis, there have been attempts to target mitochondria to treat cancer, but these attempts have had limited results. One treatment possibility related to cancer therapy is mitochondrial infusion. Mitochondria are first isolated from cell cultures or tissue (Figure 1A). Donor and recipient cells are plated for 24 h, and the mitochondria of the two cell lines are stained with two distinct colors on Day 2; on Day 3, the donor mitochondria are isolated and infused into the recipient cells, so on Day 4, uptake can be visualized [28]. Figure 1B shows the protocol to isolate mitochondria from mice. Mouse liver is excised and homogenized with a dissociator, and the mitochondria are isolated via filtration and centrifuge. Mitochondria are infused into the cells or mice within 1 hour of isolation. Uptake can be visualized the next day because the mitochondria in the mouse liver are labeled green fluorescent protein (GFP) [29]. For in vitro studies, cells are plated 1 day before mitochondrial isolation.

### 1.5. The “Good” of Mitochondrial Infusion

Mitochondrial infusions have reduced ischemic injury in liver and kidney mouse models by improving respiration and ATP production [29,30]. Regarding cancer, when healthy mouse liver mitochondria were injected into melanoma mice models with lung metastasis, there was an increase in survival, a decrease in metastasis to the lungs, a decrease in ATP production, and an increase in p53 [31]. The elevation in p53 suggests that more apoptosis is occurring; when compared to the ischemic injuries, this study used mitochondria to kill the cancerous cells by depriving them of energy as seen with the decrease in ATP. Mitochondria from younger mice increased ROS and decreased ATP to a greater degree [31]. The difference in oxygen consumption being greater in younger mitochondria, but it may be due to the accumulation of mutations in older mitochondria; as DNA replicates, errors are inevitable, so genes may become inactivated or altered. Infusing healthy mitochondria into TNBC may help with apoptosis by providing functional proapoptotic proteins and possibly producing more ROS to induce further damage and apoptosis.

Several studies have looked at the effect of mitochondrial infusion in mice and human cell lines. Adding healthy mitochondria to 143B TK (human bone osteosarcoma) cell cultures reduced tumorigenesis, such as proliferation, survival, and resistance; ATP synthesis also improved [32]. Mitochondria may be inducing apoptosis, which can be due to the presence of functional proapoptotic genes in healthy mitochondria. Furthermore, when mitochondria were infused into U87 (human glioma cell line) prior to radiation therapy, there was an increase in cytochrome c and cleaved caspase 9, suggesting that apoptosis was occurring; researchers concluded that mitochondrial infusion improved radiosensitivity [33]. The preceding study also noted a decrease in hexokinase 2 (HK2) and pyruvate kinase 2, while an increase in citrate synthase (CS) and isocitrate dehydrogenase 2 (IDH2), suggesting a decrease in glycolysis and an increase in OXPHOS [33].

When mitochondria were infused into MDA-MB-231 and MCF7 breast cancer cell lines, there was increased apoptosis, a decrease in proliferation, and a decrease in ROS; apoptosis-inducing factor (AIF) translocated to the nucleus to induce apoptosis by causing chromosomal condensation and fragmentation [34]. In human TNBC, mitochondrial infusion with doxorubicin or paclitaxel decreased cell viability [34]. Furthermore, there was a decrease in several glycolytic enzymes in MDA-MB-231 after mitochondrial infusion, suggesting less reliance on glycolysis after mitochondrial infusion [35]. However, the researchers did not look at the Citric Acid Cycle to see if OXPHOS was occurring. In a different study with MDA-MB-231 and MCF7, there was a decrease in mito-ERꞵ in TNBC with a decrease in ATP production, but upregulating mito-ERꞵ activated OXPHOS and inhibited growth; there was an increase in various subunits of the NADH dehydrogenase, various cytochrome c oxidase, and ATPase6/8, suggesting less reliance on glycolysis and more on OXPHOS [36].

A few studies have examined the effect of mitochondrial infusion on breast cancer mouse models. In one study, MDA-MB-231 cells were injected into mice; when the mice were treated with mitochondria, there was a decrease in tumor weight/volume, decrease in proliferation, and increase in apoptosis; there was also an increase in fusion proteins, mitochondrial dynamin-like GTPase (OPA1) and mitofusin 2 (MFN2) and a reduction in fission protein dynamin-related protein 1 (DRP1) [37]. In a different study, DRP1 inhibition improved cisplatin-induced mitophagy and apoptosis in hepatocellular carcinoma cells by increasing BAX [38]. Mitophagy is the selective autophagy of damaged mitochondria. On the other hand, mitophagy can be hindered with BRCA1 deficiency [21].

There have been many mitochondrial infusions in humans since 2017. The first infusions were in 5 pediatric patients, ages 2 days old to 2 years old, on extracorporeal membrane oxygenation (ECMO) for ischemic reperfusion dysfunction status-post cardiac surgery; when mitochondria were isolated from the rectus abdominis and injected directly into the heart, four of the patients improved and were successfully taken off ECMO [39]. A similar study was published in 2020, in which 12 of the 24 pediatric patients on ECMO for ischemic reperfusion injury were given mitochondria; 8 of the 12 patients were taken off ECMO and showed ventricular improvements [40]. Even though these are not studies related to cancer and mitochondrial infusion, these two studies provide optimism for future mitochondrial infusions in many diseases, including cancer.

### 1.6. The “Bad” of Mitochondrial Infusion

Even though there have been many studies that show a positive impact of mitochondrial infusion, there have also been studies that show increased tumorigenesis. One issue in this method has been the biochemical composition of the mitochondrial membrane and the strong negative membrane potential making delivery of drugs challenging.

Mitochondrial infusions from stem cells to human breast cancer cells have been shown to increase proliferation and invasion [28], in contrast to the findings above. This suggests that the source of the mitochondria (stem cells vs. liver or other tissue) may alter the effects. When mitochondria were transferred from mesenchymal stem cells to neuronal stem cells, there was a decrease in apoptosis after cisplatin treatment [41]. Cisplatin typically reduces the mitochondrial membrane potential, but the preceding study suggests that it can be reversed with mitochondrial transfer [41]. PC12 (pheochromocytoma) cells survived ultraviolet light treatment when they were able to acquire mitochondria from untreated cells [42].

Mitochondria can also promote tumorigenesis in some cancer cell lines. In B16p (mouse melanoma) and 4T1 (mouse TNBC) cell cultures without mtDNA, acquisition of mtDNA improved visible cristae and measurable respiratory function, leading to increased tumorigenesis; there was an increase in succinate dehydrogenase A (SDHA) [43,44]. Furthermore, when MCF7 cells acquired mitochondria via nanotubes from stromal or epithelial cells, the cancer line became resistant to chemotherapy [45]. When hormonal therapy resistant metastatic breast cancer cells were grafted into mice, there was an uptake of murine mtDNA; these cells underwent more OXPHOS, hence playing a role in proliferation [45]. Nanotubes have also been shown to cause chemoresistance in Jarkat cells (immortalized human T lymphocyte cell line) through the expression of intracellular adhesion molecule 1 (ICAM-1); in the presence of an anti-ICAM-1 antibody, there was a reduction in mitochondrial transfer and improvement in treatment sensitivity [46]. Even though mitochondrial infusions seem to be beneficial to treating many diseases, it is important to consider the downfalls of such as well.

## 2. Conclusions

### The Past, Present, and Future

TNBC affects thousands of women each year, so research into targeted treatments is important due to the lack of specific therapies available. The development of such therapies has been challenging due to the incidence of resistance as well as treatment failure. The current most common treatment for TNBC is combination chemotherapy followed by surgery, but five-year survival rates are still significantly lower when compared to other forms of breast cancer. Cancer, including TNBC, has been shown to undergo the Warburg Effect and increase energy metabolism even with reduced perfusion or necessary oxygen supply. There is evidence to suggest that mitochondria have a significant role in not only the progression but the metastatic potential of cancer. Mitochondria are important in initiating apoptosis in abnormal cells and resistance to apoptosis is one mechanism of the development of neoplasms. This resistance can come from abnormal mitochondria or altered apoptotic genes. Mitochondria are also a major source of ROS. A central inflammatory niche with constitutive elevations of ROS has been shown to drive tumor progression and enhance metastasis. Alternatively, there seems to be a toxic threshold of ROS where they no longer cause cancer progression but instead apoptosis, a Goldilocks effect for ROS signaling cancer cell survival or death.

Mitochondria’s role in cancer has caused some to consider mitochondria infusion as a possible treatment choice for TNBC. Earlier studies have documented successful mitochondria infusion, and some have also shown decreased tumor weight as well as reduced metastasis and increased survival in mouse models. This finding is possibly due to an increase in apoptosis by the infused healthy mitochondria, shown by elevated p53. Some research, however, has shown an increase in tumorigenesis with mitochondria infusion. The complex relationship between mitochondria and cancer progression along with conflicting research related to mitochondrial infusion as a potential targeted treatment warrants continued investigation. As TNBC is an aggressive and often fatal malignancy affecting many women every year, investigation into potential targeted treatments is of significant importance as responses to standard treatments up to this point have been limited.

## Figures and Tables

**Figure 1 ijms-23-12993-f001:**
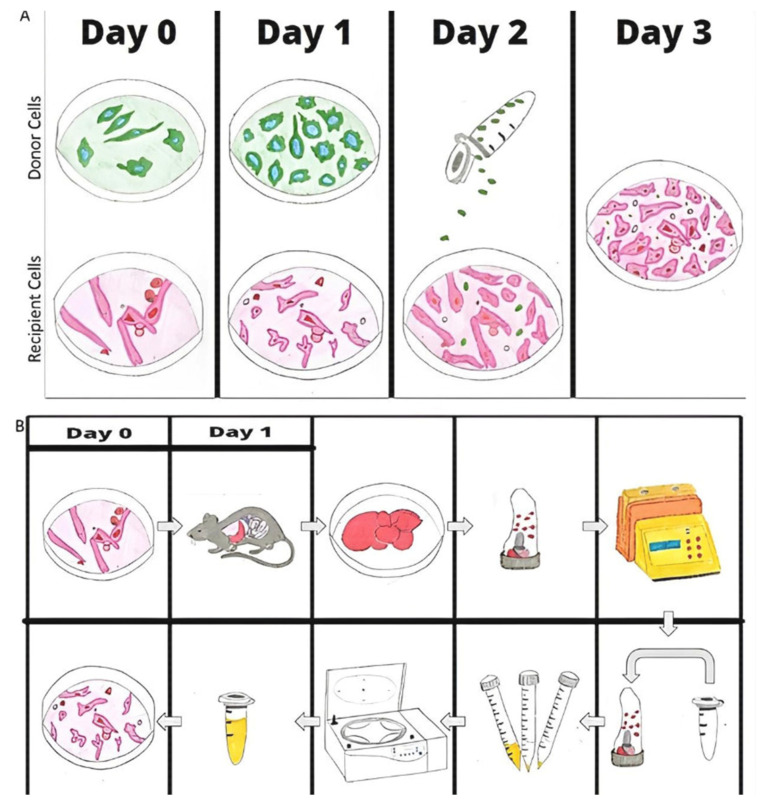
Mitochondrial Isolation: Basic protocol of mitochondrial isolation from cells (**A**) and tissue from mice (**B**). A. donor and recipient cells plated and stained with 2 distinct colors; the donor mitochondria are isolated and infused into the recipient cells. Mitochondria are isolated from mouse liver via dissociation and centrifuge. Mitochondria are infused into the cells or mice within 1 hour of isolation. Uptake can be visualized the next day because the mitochondria in the mouse liver are labeled with GFP.

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
