# Peer review of "Mitochondrial Role in Oncogenesis and Potential Chemotherapeutic Strategy of Mitochondrial Infusion in Breast Cancer"

_ijms, 2022, doi:10.3390/ijms232112993_

Round 1

Reviewer 1 Report

It deserves several comments:
-    Major comments : The first part is not adapted to the subject indicated in the title. It is not very interesting, too long and too general. This is the case of epidemiology, tumorigenesis and cisplatin which is out of the question.
The section on the role of mitochondria in oncogenesis and/or treatment resistance is incomplete and could be updated.
The real novelty of this review is the mitochondrial infusion. The fact that there is a natural infusion is hardly mentioned, but the transfer of mitochondria between normal and cancer cells has been well described in many models and its mechanisms have been described.
-    minor points: poor quality of Figure 3, the abstract is not well written and only partially corresponds to the title

Author Response

Reviewer 1

Thank you for the comments. We agree with each of the comments, so we have made the changes in blue and red in the document and have also indicated below.

It deserves several comments:
-    Major comments: The first part is not adapted to the subject indicated in the title. It is not very interesting, too long and too general. This is the case of epidemiology, tumorigenesis and cisplatin which is out of the question.

  • Removed Hallmark of Cancer paragraph and figure.
  • Rearranged and condensed 1.3 through 1.5 into one new section (1.3).
  • Removed 1.6, which was about cisplatin and the associated figure.
  • Changed title.

The section on the role of mitochondria in oncogenesis and/or treatment resistance is incomplete and could be updated.

  • Added information to 1.3 and 1.6. New resources included.

The real novelty of this review is the mitochondrial infusion. The fact that there is a natural infusion is hardly mentioned, but the transfer of mitochondria between normal and cancer cells has been well described in many models and its mechanisms have been described.

  • Added information to 1.4. Discussed mechanisms of natural infusions.

-    minor points: poor quality of Figure 3, the abstract is not well written and only partially corresponds to the title

  • Adjusted the figure. Figure 1 now since the other 2 were removed.

Reviewer 2 Report

This review recapitulates the impact of mitochondria in cancer in terms of metabolism and type of cell death. Here are my comments:

-        Breast cancer is detailed but by reading the title we have the idea that it’s cancer in general which will be discussed. Title should then be more focused.

-        What about ferroptosis in which mitochondria has been shown to play a role – should be added in the ROS paragraph.

-        Minor typos can be found and should be corrected.

-        Line 273, M.P. Lisanti’s work should be added and discussed – his work on mitochondria in cancer has indeed paved the way for many researchers in this field and discussing his work would strengthen this review.

-        Some seminal references are lacking and should definitely be added:

o   Porporato PE, Filigheddu N, Pedro JMB, Kroemer G, Galluzzi L. Mitochondrial metabolism and cancer. Cell Res. 2018 Mar;28(3):265-280. doi: 10.1038/cr.2017.155. Epub 2017 Dec 8. PMID: 29219147; PMCID: PMC5835768.

o   DeBerardinis RJ, Chandel NS. We need to talk about the Warburg effect. Nat Metab. 2020 Feb;2(2):127-129. doi: 10.1038/s42255-020-0172-2. PMID: 32694689.

o   Cassim S, Pouyssegur J. Tumor Microenvironment: A Metabolic Player that Shapes the Immune Response. Int J Mol Sci. 2019 Dec 25;21(1):157. doi: 10.3390/ijms21010157. PMID: 31881671; PMCID: PMC6982275.

o   DeBerardinis RJ, Chandel NS. Fundamentals of cancer metabolism. Sci Adv. 2016 May 27;2(5):e1600200. doi: 10.1126/sciadv.1600200. PMID: 27386546; PMCID: PMC4928883.

o   Cassim S, Vučetić M, Ždralević M, Pouyssegur J. Warburg and Beyond: The Power of Mitochondrial Metabolism to Collaborate or Replace Fermentative Glycolysis in Cancer. Cancers (Basel). 2020 Apr 30;12(5):1119. doi: 10.3390/cancers12051119. PMID: 32365833; PMCID: PMC7281550.

Author Response

Reviewer 2

This review recapitulates the impact of mitochondria in cancer in terms of metabolism and type of cell death. Here are my comments:

-        Breast cancer is detailed but by reading the title we have the idea that it’s cancer in general which will be discussed. Title should then be more focused.

  • Changed title.

-        What about ferroptosis in which mitochondria has been shown to play a role – should be added in the ROS paragraph.

  • ROS section has been moved under 1.3. Added information to 1.3.

-        Minor typos can be found and should be corrected.

  • Addressed grammar issues.

-        Line 273, M.P. Lisanti’s work should be added and discussed – his work on mitochondria in cancer has indeed paved the way for many researchers in this field and discussing his work would strengthen this review.

  • Added to 1.2. Discussed Reverse Warburg Effect and included studies.

-        Some seminal references are lacking and should definitely be added:

 o   Porporato PE, Filigheddu N, Pedro JMB, Kroemer G, Galluzzi L. Mitochondrial metabolism and cancer. Cell Res. 2018 Mar;28(3):265-280. doi: 10.1038/cr.2017.155. Epub 2017 Dec 8. PMID: 29219147; PMCID: PMC5835768.

  • Added to 1.3

o   DeBerardinis RJ, Chandel NS. We need to talk about the Warburg effect. Nat Metab. 2020 Feb;2(2):127-129. doi: 10.1038/s42255-020-0172-2. PMID: 32694689.

  • Not included

o   Cassim S, Pouyssegur J. Tumor Microenvironment: A Metabolic Player that Shapes the Immune Response. Int J Mol Sci. 2019 Dec 25;21(1):157. doi: 10.3390/ijms21010157. PMID: 31881671; PMCID: PMC6982275.

  • Added to 1.3

o   DeBerardinis RJ, Chandel NS. Fundamentals of cancer metabolism. Sci Adv. 2016 May 27;2(5):e1600200. doi: 10.1126/sciadv.1600200. PMID: 27386546; PMCID: PMC4928883.

  • Added to 1.2

o   Cassim S, Vučetić M, Ždralević M, Pouyssegur J. Warburg and Beyond: The Power of Mitochondrial Metabolism to Collaborate or Replace Fermentative Glycolysis in Cancer. Cancers (Basel). 2020 Apr 30;12(5):1119. doi: 10.3390/cancers12051119. PMID: 32365833; PMCID: PMC7281550.

  • Added to 1.3

Round 2

Reviewer 1 Report

The authors have taken into account some of my comments and have improved the manuscript.

There are still unacceptable imperfections:

-          - line 90 and 91: it is not MOMP/MPT that is responsible for tumor formation but rather its blockage that makes cancer cells intrinsically resistant to cell death

-          - the bibliography is incomplete with many invalid citations

Author Response

The authors have taken into account some of my comments and have improved the manuscript.

Thank you for the comments. We agree with each of the comments and apologize for the imperfections in our previous submission.  I (AB) take all ownership of the mistakes and appreciate your time and thankful for the opportunity to address these concerns.  I failed to correct them in my previous submission after the students had worked hard to fix them.  Thank you again and hope we have corrected all mistakes.

There are still unacceptable imperfections:

-          - line 90 and 91: it is not MOMP/MPT that is responsible for tumor formation but rather its blockage that makes cancer cells intrinsically resistant to cell death

  • Corrected the wording in Lines 90 and 91.

-          - the bibliography is incomplete with many invalid citations

  • Apologies, we have gone through each citation individually and verified and update accordingly.

Round 3

Reviewer 1 Report

The manuscript has been improved following my remarks

Author Response

Thank you!